# Long-Term Functional Hyperemia after Uncomplicated Phacoemulsification: Benefits beyond Restoring Vision

**DOI:** 10.3390/diagnostics12102449

**Published:** 2022-10-10

**Authors:** Ana Ćurić, Mirjana Bjeloš, Mladen Bušić, Biljana Kuzmanović Elabjer, Benedict Rak, Nenad Vukojević

**Affiliations:** 1University Eye Department, Reference Centre of the Ministry of Health of the Republic of Croatia for Pediatric Ophthalmology and Strabismus, University Hospital “Sveti Duh”, Sveti Duh 64, 10000 Zagreb, Croatia; 2Faculty of Dental Medicine and Health Osijek, Josip Juraj Strossmayer University of Osijek, 31000 Osijek, Croatia; 3Faculty of Medicine, Josip Juraj Strossmayer University of Osijek, 31000 Osijek, Croatia; 4Department of Ophthalmology, University Hospital Centre Zagreb, 10000 Zagreb, Croatia; 5School of Medicine, University of Zagreb, 10000 Zagreb, Croatia

**Keywords:** cataract, retinal vessels, hyperemia, phacoemulsification, intraocular pressure

## Abstract

The purpose of the study was to investigate the long-term effects of uncomplicated phacoemulsification on macular perfusion using optical coherence tomography angiography (OCTA) in healthy aging subjects. OCTA was performed before phacoemulsification and 1 week, 1 month, 3 months, and 6 months after. Superficial vascular complex (formed of nerve fiber layer vascular plexus and superficial vascular plexus), deep vascular complex (formed of intermediate capillary plexus and deep capillary plexus), as well as choriocapillaris (CC) and large choroidal blood vessels were recorded. Significant changes of vascular parameters in 95 eyes of 95 patients reached plateau 1 week after surgery and remained stable up to 6 months, occurring in all retinal layers but not in choroid and CC. Statistically significant increases in retinal vessels area, vessels percentage area, total number of junctions, junctions density, and total and average vessels length were found, followed by the total number of end points and mean lacunarity decline, proving an increase in blood supply. The study confirmed that uncomplicated phacoemulsification leads to a long-term increase in macular retinal perfusion. The results might ease the decision regarding timing for cataract surgery as long-term perfusion benefits can be achieved. Furthermore, study results provide a normative database of retinal and choroidal vasculature in healthy aging patients.

## 1. Introduction

The vascular system of the macula, as the area responsible for the clearest vision, is crucial for normal eye function [1]. The effects of phacoemulsification on macular blood vessels and their duration are still debatable [1,2,3]. In the previous studies, an increase in macular thickness after uncomplicated phacoemulsification was related to local inflammatory response [2,3], although the direct evidence of inflammation was not provided. Opposed to this unfavorable inflammatory effect ceasing after one to three months [2,4], the beneficial phenomenon of functional hyperemia has been suggested recently as the cause of the retinal microvasculature changes [5].

Functional hyperemia, also known as neurovascular coupling, is a highly sophisticated mechanism conjoining neuronal functional activity and blood flow [6]. This mechanism is thought to be involved in the development of glaucoma, diabetes, and systemic hypertension as proved by the decline in vessel diameter following full-field stimulation with flickering light prior to clinically detectable retinal damage [6].

The aim of this study was to investigate the long-term effects of uncomplicated phacoemulsification on macular perfusion on a large sample of patients with incipient and immature senile cataracts. The current study takes advantage of recently provided evidence demonstrating three-month increase in macular perfusion after uncomplicated phacoemulsification, most likely due to functional hyperemia evoked by increased light intensity stimulation of the retina after cataract removal [5]. 

## 2. Materials and Methods

The prospective study on one group of patients was conducted at the University Eye Department, University Hospital “Sveti Duh”, Zagreb. All analyzed parameters refer to one eye. The research was approved by the Ethics Committee of the University Hospital “Sveti Duh” Zagreb and the Ethics Committee of the University of Zagreb, School of Medicine (protocol code 01-4212/2, 24 February 2020). The research adhered to the tenets of the Declaration of Helsinki. All participants signed informed consent prior to enrollment.

Inclusion criteria for the study were: senile cataract that is not complicated by other ophthalmic diseases; optical coherence tomography angiography (OCTA) image quality index (QI) ≥ 30 calculated with integrated HRA + OCT Spectralis^®^ (Heidelberg Engineering, Heidelberg, Germany) software version 6.16.7; cataract classified according to the Pentacam^®^ nucleus staging (PNS) system as 1, 2 or 3, and the Lens Opacities Classification System III (LOCS III) classification as NO 0–3, NC 0–3, C 0–3, P 0–3; axial length (AL) of the eyeball measured by optical biometrics 20–25 mm; intraocular pressure (IOP) value measured by Goldmann applanation 10–21 mmHg; blood pressure value ≥ 90/60 mmHg and ≤140/90 mmHg.

Exclusion criteria were: type 1 or 2 diabetes; corneal diseases; pseudoexfoliative syndrome; glaucoma; posterior eye segment and retinal disorders that could affect OCT and OCTA measurements; childhood, juvenile and traumatic cataracts; intraoperative and/or postoperative complications; unregulated hypertension.

Preoperative examination included: history taking, determining the best corrected visual acuity (BCVA), autorefractokeratometry, endothelial biomicroscopy, optical biometry, IOP measurement, fundus examination in mydriasis, OCT and OCTA of the macula, and arterial pressure measurement.

BCVA was measured at 4 m using the ETDRS table and recorded in the logMAR unit. Corneal endothelial cells were analyzed using endothelial biomicroscope (CEM-530, Nidek, Tokyo, Japan). AL of the eye was measured using optical biometer (IOLMaster^®^ 700, Zeiss, Oberkochen, Germany), and the intraocular lens (IOL) power (AMO Tecnis PCB00, Johnson & Johnson Vision, Jacksonville, Florida, USA) was further computed with SRK-T formula targeting emmetropia. Arterial pressure was read using a digital blood pressure monitor (M6 Comfort, Omron, Kyoto, Japan). IOP was measured by Goldmann applanation tonometry. Mean arterial pressure (MAP) was determined as: [systolic blood pressure (SBP) + 2× diastolic blood pressure (DBP)]/3. Ocular perfusion pressure (OPP) was figured according to the formula: 2/3 (SAP-IOP).

Tropicamide 1% (Mydriacyl^®^, Alcon Laboratories Inc., Geneva, Switzerland) was instilled into each eye of the patient three times at intervals of 15 min to achieve the best possible mydriasis and cycloplegia. Examination of the anterior segment of the eye and determination of lens opacity according to LOCS III classification was performed using a slit lamp, fundus examination using an indirect ophthalmoscope, measurement of objective refraction using autorefractokeratometer (Righton Speedy-K, Nidek, Tokyo, Japan), and PNS classification using Pentacam^®^ HR (OCULUS Optikgeräte GmbH, Wetzlar, Germany).

OCT, OCTA and fundus autofluorescence were recorded using HRA + OCT Spectralis^®^ (Heidelberg Engineering, Heidelberg, Germany). OCTA recorded 10° × 10° (2.9 mm × 2.9 mm) central part of the macula using 512 A-scans × 512 cross-sections, with a distance of 6 µm between sections and a resolution of 5.7 µm per pixel. Superficial layer (nerve fiber layer vascular plexus (NFLVP) and superficial vascular plexus (SVP) that form superficial vascular complex (SVC)) and deep layer (deep capillary plexus (DCP) and intermediate capillary plexus (ICP) that form deep vascular complex (DVC)) of retinal blood vessels, as well as large choroidal vessels and choriocapillaris (CC) were recorded. Automatic layer segmentation was made by integrated Spectralis^®^ software, considered reliable in cases without posterior segment eye pathology [7]. The recording was performed in high resolution of low measurement variability [8]. In case of artifacts, the measurement was repeated. QI of each section was calculated using integrated HRA + OCT Spectralis^®^ software version 6.16.7 Images with QI ≥ 30 were further analyzed and carefully reviewed by two independent examiners for accurate segmentation of vascular layers and the presence of artifacts prior to final analysis [7]. Images with QI < 30 were not further analyzed, and those patients were not included in the study.

Images were exported to AngioTool^®^ 0.6 software (National Institutes of Health, National Cancer Institute, Bethesda, MD, USA) for quantitative analysis of blood vessels [9]. After image segmentation, a blood vessel was defined as the segment between two branching points or branching point and an end point [9].

Analyzed vascular parameters included: explant area (EA), analyzed area; vessels area (VA), area of segmented blood vessels; vessels percentage area (VPA), the percentage of the area containing segmented blood vessels within the explant area (VA/EA); total number of junctions (TNJ), total number of vessels junctions in the image; junctions’ density (JD), number of vessel junctions per unit area (branch points/unit area); total vessels length (TVL), the sum of Euclidean distances between the pixels of all blood vessels in the figure; average vessels length (AVL), mean length of all vessels in the image; total number of end points (TNEP), number of open-ended segments; mean lacunarity (ML), mean lacunarity overall size boxes; and foveal avascular zone (FAZ) surface.

FAZ surface was manually marked after segmentation by one examiner (A.Ć.) using Adobe Photoshop CS6 64 Bit software (Adobe Inc., San Jose, CA, USA). The internal boundaries of blood vessels were interconnected to form the external boundaries of FAZ. After that, the software automatically calculated FAZ surface in mm^2^.

FAZ surface was manually marked after segmentation by one examiner (A.Ć.) using Adobe Photoshop CS6 64 Bit software (Adobe Inc., San Jose, CA, USA). The internal boundaries of blood vessels were interconnected to form the external boundaries of FAZ. After that, the software automatically calculated FAZ surface in mm^2^.

All subjects underwent microincisional cataract surgery under topical anesthesia performed by an experienced surgeon (B.K.E.) using the Centurion^®^ Vision System (Alcon Inc., Fort Worth, TX, USA). Total cumulative dissipated energy (CDE) and total ultrasound time (PHACO time) were automatically recorded. Foldable intraocular lens (AMO Tecnis PCB00) was implanted. Dexamethasone 0.1% drops (Maxidex^®^, Alcon Laboratories Inc., Geneva, Switzerland) were prescribed postoperatively q.i.d. for the first 7 days, followed by b.i.d. for another 7 days.

Subjects were evaluated before surgery and 1 week, 1 month, 3 months, and 6 months postoperatively. At the follow-up examinations, HRA + OCT Spectralis^®^ follow-up protocol was selected. The same procedures were performed at follow-up examinations as at the preoperative examination. All subjects were examined and recorded by one examiner (A.Ć.), who also analyzed all parameters.

Statistical analysis was performed using MedCalc statistical software (MedCalc Software Ltd., Ostend, Belgium). Variables were presented as median and interquartile ranges. Comparison of the values of general and vascular parameters before phacoemulsification with the values obtained 1 week, 1 month, 3 months, and 6 months after surgery was performed using the nonparametric Friedman ANOVA test. The significance level was set at *p* < 0.001.

## 3. Results

Of 100 subjects who met the inclusion criteria, 95 eyes of 95 patients were included in the final analysis. One patient developed postoperative iridocyclitis and was excluded from further analysis. Three patients missed follow-up examinations. One patient developed pseudophakic macular edema diagnosed 1 month after phacoemulsification using OCT, with no signs of edema 3 months after surgery, but was also excluded from the final analysis. General characteristics of the subjects and parameters recorded during phacoemulsification are shown in Table 1.

### 3.1. General Parameters

IOP before phacoemulsification was significantly higher than 1 week, 1 month, 3 months, and 6 months after, among which there were no statistically significant differences. BCVA improved significantly after surgery. No statistically significant differences were found in the parameters SBP, DBP, MAP, and OPP (Table 2).

### 3.2. Vascular Parameters

No statistically significant difference was found in EA. A statistically significant increase in VA and VPA was found in all layers except in CC, while in choroid a decrease was found. An increase in JD was found in all layers except CC. TNJ, TVL, and AVL increased significantly in all retinal layers, but no difference was found in choroid and CC. TNEP significantly decreased in all layers except choroid and NFLVP, while ML significantly decreased in all layers except choroid and CC (Table 3 and Table 4). 

The most pronounced changes in vascular parameters were found in NFLVP and SVC (Table 4). AVL is the parameter with the largest changes in all layers except in NFLVP (Table 4). For all vascular parameters, a statistically significant difference was demonstrated 1 week, 1 month, 3 months, and 6 months after phacoemulsification, within which there were no statistically significant differences.

### 3.3. Nerve Fiber Layer Vascular Plexus, Superficial Vascular Plexus, and Superficial Vascular Complex

In NFLVP, vascular parameters: VA, VPA, TNJ, JD, TVL, and AVL before phacoemulsification were significantly lower than 1 week, 1 month, 3 months, and 6 months after, within which no statistically significant differences were found (Appendix A). No significant changes were found for TNEP, while ML was significantly higher before than after phacoemulsification. The largest changes were found in TNJ and JD (63.43% and 62.91%) (Appendix A).

In SVP and SVC, vascular parameters: VA, VPA, TNJ, JD, TVL, and AVL before phacoemulsification were significantly lower than 1 week, 1 month, 3 months, and 6 months after phacoemulsification within which no statistically significant differences were found. TNEP and ML were significantly higher before phacoemulsification than 1 week, 1 month, 3 months and 6 months after, within which no statistically significant differences were found (Appendix A). AVL showed the most pronounced changes (53.13% and 165.76%) (Appendix A).

### 3.4. Intermediate Capillary Plexus, Deep Capillary Plexus and Deep Vascular Complex

In ICP and DCP, as well as in DVC, 1 week after phacoemulsification a statistically significant increase in VA, VPA, TNJ, JD, TVL and AVL was found together with a decrease in TNEP and ML (Appendix A). No statistically significant difference was found among the values 1 week, 1 month, 3 months, and 6 months after phacoemulsification for any parameter. Furthermore, TNEP and ML were significantly higher in all layers before phacoemulsification compared to 1 week, 1 month, 3 months and 6 months after, within which no statistically significant differences were found. The largest change was recorded for AVL in all layers (93.70%, 88.72%, and 184.83%) (Appendix A).

### 3.5. Retina

Observing the entire retina, 1 week after phacoemulsification, an increase in parameters: VA, VPA, TNJ, JD, TVL, and AVL, and a decrease in TNEP and ML was found (Appendix A). No statistically significant difference was found among the values 1 week, 1 month, 3 months, and 6 months after phacoemulsification for any parameter. The largest change was recorded for AVL (186.04%) (Appendix A).

### 3.6. Choriocapillaris

In CC, parameter TNEP before phacoemulsification had higher values than 1 week, 1 month, 3 months, and 6 months after surgery, without significant differences among them (Appendix A).

### 3.7. Choroid

In medium-sized and large choroidal vessels, VA and VPA were significantly higher before phacoemulsification than 1 week, 1 month, 3 months, and 6 months after, with no statistically significant differences within them. Parameters JD and TNEP were significantly lower before phacoemulsification than 1 week, 1 month, 3 months, and 6 months after, without statistically significant differences within them. No significant differences were found in other vascular parameters (Appendix A).

### 3.8. Quality Index

Average QI of OCT-A images before and 1 week, 1 month, 3 months, and 6 months after phacoemulsification showed no statistically significant differences (Table 5).

### 3.9. Foveal Avascular Zone

FAZ surface was significantly larger before phacoemulsification than 1 week, 1 month, 3 months, and 6 months after, among which no significant differences were found (Table 6).

## 4. Discussion

### 4.1. Major Results

In this study, vascular parameters: VA, VPA, TNJ, JD, TVL, AVL, TNEP, and ML, throughout SVC (NFLVP and SVP) and DVC (ICP and DCP) of retinal blood vessels, as well as large and medium-sized choroidal vessels and CC were monitored. The results provided further evidence that uncomplicated phacoemulsification causes significant, sustained 6-month increase in macular perfusion. We consider these changes to be caused by functional hyperemia.

Statistically significant increase in retinal VA, VPA, TNJ, JD, TVL, and AVL, followed by significant TNJ and ML decline, proved an increase in blood supply of the central 10° × 10° (2.9 mm × 2.9 mm) of the macula after phacoemulsification (Table 3 and Table 4). In addition, a reduction in FAZ surface was found (Table 6). Changes in vascular parameters occurred as early as 1 week after phacoemulsification and remained stable for 6 months. The changes affected all retinal vascular layers, but not choroid and CC in the same extent and in the same way (Table 3 and Table 4).

### 4.2. Causes of Retinal Increased Perfusion

This study confirmed in a new, larger sample of patients that the increase in blood flow after uneventful phacoemulsification is due to functional hyperemia in retinal blood vessels caused by increased stimulation of the retina with light after cataract removal [5] and not inflammation, as previously suggested [2,3]. The study evidenced that increase in macular perfusion persisted 6 months after phacoemulsification, when the inflammatory response should no longer be present [2,4,10]. In this study, only one patient developed subclinical pseudophakic macular edema, detected 1 month after phacoemulsification. The sample of only one patient was too small for further statistical analysis. 

Previous research has demonstrated significant increase in retinal perfusion at 1 and 3 months postoperatively [2,3,5,11,12,13], some owing to the effect to IOP lowering and the extension of the anterior chamber [11,12,13]. Decrease in IOP could not be attributed as presumed reason for the increase in macular perfusion due to OPP consistency (Table 2). 

The observed differences between retina and choroid in this study were somewhat expected. Choroid demonstrated no changes in perfusion, further supporting the mechanism of functional hyperemia, recognized as an essential component of blood flow regulation present only in the retina [14,15]. In the retina, the increase in metabolism due to light stimulation lowers O_2_ and glucose levels and leads to production of vasoactive metabolites [16]. The products of neuronal activity adenosine, lactate, and arachidonic acid cause vasodilatation and functional hyperemia to compensate for energy consumption and increase in neuronal activity restoring O_2_ and glucose levels [14,17,18]. Thus, when neurons are activated by light stimulation, there is a substantial increase in retinal blood flow by over 59%, generated by active dilation of retinal arteries, arterioles, and capillaries [19]. Opposed to the retina, light stimulation has a little effect on choroidal circulation and is insensitive to pO_2_ fluctuations [17].

### 4.3. Retina

In humans, when the retina is activated by light stimulation, there is an increase in basal blood flow with both primary artery and vein dilatation of 3–7% [20]. We hypothesize the hyperemia causes greater tortuosity of network with further opening of nonfunctional capillary segments as indicated by AVL and JD increase (Table 3 and Table 4). In contrast to the above parameters, TNEP and ML decreased significantly after phacoemulsification (Table 3 and Table 4). TNEP is a parameter associated with the total number of blind terminal capillaries [9], so a decrease in the TNEP would speak in favor of anastomotic opening. Lacunarity measures the distribution of gaps between blood vessels and is an indicator of vascular structural inequality [9]. ML decline is beneficial event related to intercapillary areas narrowing and oxygen diffusion time decrement [9]. As a consequence of physiological aging and altered visual stimulation (i.e., cataract), decreased metabolic activity disrupted the homogeneity of DCP and ICP more than SVP, where capillary dropout was more common as presented with larger capillary recruitment in Appendix A. These differences confirmed that the inner segments of the photoreceptors together with the inner and outer plexiform layers (IPL and OPL) are the areas with the highest oxygen consumption. Such a pattern was supported by Haddad et al., who found a significant changes in SVP but even more profound changes in DVP [20]. 

Recent OCTA studies addressing macular perfusion after uncomplicated phacoemulsification reached conflicting results [11,21,22]. Although conducted on a relatively large number of subjects (N = 107, N = 58, and N = 44), cataract density was assessed subjectively [11,21] or was not assessed at all [22], different OCTA devices were used, no analysis of QI and EA parameters before and after phacoemulsification was performed, while the subjects had lower preoperative BCVA compared to our study. The above settings could all be raised as potential sources of bias that may significantly influence quantitative measurements and confound the outcomes.

### 4.4. Choroid and Choriocapillaris

Changes in choroid and CC did not follow the patterns of changes found in retinal blood vessels (Table 3, Appendix A). In CC, only TNEP altered significantly (Appendix A). The decrease in TNEP could indicate anastomotic opening and reflow due to increased heat dissipation [23]. 

The major inconsistencies in the data exist in choroidal response after phacoemulsification [10,24,25,26]. The changing rate of the parameters VA, VPA, JD, and TNEP, although statistically significant, seemed of low magnitude and suggested constriction following the TNEP recruitment in CC. However, further studies should elucidate whether these alterations are of clinical importance as light stimulation and pO_2_ fluctuations have little effect on choroidal circulation, as opposed to retinal blood vessels [17]. Earlier results proved that changes in the retina after uncomplicated phacoemulsification were not accompanied by changes in the choroid [2,5,10,26], casting further doubt on the clinical importance of the aforementioned statistically significant results.

In view of the above, this study confirmed that physiological requirements under uncomplicated phacoemulsification have not reached the threshold for causing the breakdown of the external blood–retinal barrier and the emergence of an inflammatory response.

### 4.5. Foveal Avascular Zone

As recommended by Campbell et al. [27], FAZ area was measured on *en-face* images showing all layers of the retina simultaneously. Previous studies have shown that increase in the diameter and surface of FAZ could predict the progression of microvascular retinal diseases [19]. In our study, FAZ surface decreased significantly, confirming the increase in macular perfusion (Table 6) and highlighting the inverse correlation between higher retinal metabolic requirements and lower FAZ. A larger area of FAZ has been associated with the severity of capillary occlusion and nonperfusion in the macular region [12,27]. Furthermore, clinical studies using OCTA have shown that aging causes decreased total retinal blood flow associated with an increase in FAZ surface [1,28].

### 4.6. Normative Database of Healthy Aging Subjects

Normal aging decreases retinal vascular density [5,29], blood flow velocity, and thickness [29]. Altered hemodynamics and hypoperfusion advance the likelihood of neurodegenerative disorders, mainly in nervous tissues with high metabolic demands [30]. The retina in particular shows loss of microvasculature and thinning of the retinal nerve fiber layer and ganglion cell layer [30]. In CC, the observed decline in vascular density and the diameter of capillaries is linear with age [31]. Our study further exhibited the decline in metabolic activity during physiological aging as macular blood flow increased significantly after uncomplicated phacoemulsification.

To the best of our knowledge, Fernandez-Vigo et al. provided the largest OCTA normative database of retinal and CC microvasculature; however, it was done using swept-source OCTA [32]. They noted wide variations in the metrics, with macular vascular density positively corelating with foveal and choroidal thickness, but negatively with age, and no relation to AL and sex was revealed. The vascular density obtained in the latter study was lower compared to VPA in our study, but different OCTA devices and different analysis algorithms were used [32]. Thus, minimizing the study bias and measurement error by using spectral domain OCTA (SD-OCTA) technique carefully, our study yielded normative data of retinal and choroidal vasculature (Appendix A) in healthy aging subjects obtained with SD-OCTA (Table 1 and Table 2).

### 4.7. Image Quality

As shown by Yu et al., moderate to severe lens opacity proved to have a significant effect on image quality and quantitative analysis of retinal blood vessels [33]. The impact of incipient cataract was not investigated [33]. Our study included only patients with incipient and immature cataracts, subjectively graded with LOCS III classification but also objectively using PNS classification (Table 1). OCTA image quality quantified by the software integrated within HRA + OCT Spectralis^®^ before and after surgery showed no statistically significant differences (Table 5), excluding the possibility of image quality bias. Therefore, the observed changes in macular perfusion could not be explained by higher QI after cataract removal. Furthermore, if the increase in blood flow was only a consequence of the QI improvement, the same changes in retina as well as in choroid and CC would be expected, while the changes of FAZ surface could not be explained.

## 5. Conclusions

In this study, a new clinical question was raised: does cataract surgery have additional long-term benefits apart from improving visual acuity? The results here presented confirmed that uncomplicated phacoemulsification led to a long-term increase in macular retinal perfusion related to functional hyperemia, underlying the evidence previously reported [5]. The increase in vascular parameters was shown to occur 1 week after phacoemulsification and remained stable for up to 6 months. We concluded that increased retinal perfusion had no clinically significant effect on choroidal circulation as the threshold to cause a breakdown of the external blood–retinal barrier was not reached. Therefore, in *lieu* of scarce scientific evidence defining the optimal time for phacoemulsification [34,35], and having in mind potential risks of early cataract surgery, future studies should elucidate whether the benefits of induced functional hyperemia outweigh the risks of early cataract surgery. Further, considering the range of applicability, the study results provide normative database of retinal and choroidal vasculature in healthy aging subjects.

The development of technology that would enable recording of high-quality high-resolution OCTA images in patients with denser cataracts would potentiate further research on this topic. The results of this study may direct further research in terms of proving functional hyperemia in other states of visual axis obstruction other than lens opacity (e.g., corneal opacity, intravitreal hemorrhage, ptosis) and provide the answer to the question of whether functional hyperemia is unique to phacoemulsification or whether other surgeries removing visual axis obstructions could similarly induce it. Furthermore, additional studies are warranted to compare the same pre- and post-surgical vascular parameters in patients with underlying ophthalmologic conditions in order to determine if similar mechanisms could be observed.

## Figures and Tables

**Table 1 diagnostics-12-02449-t001:** General characteristics of the patients and parameters recorded during phacoemulsification.

Age (Years)	73 (65–77)
Sex (male, %)	M: 37/95 (38.9%)
(female, %)	F: 58/95 (61.1%)
LOCS III classification (n/N)	NO—0: 2/95 (2.1%); 1: 25/95 (26.3%); 2: 60/95 (63.2%); 3: 8/95 (8.4%)
	NC—0: 2/95 (2.1%); 1: 27/95 (28.4%); 2: 58/95 (61.1%); 3: 8/95 (8.4%)
	C—0: 37/95 (38.9%); 1: 16/95 (16.8%); 2: 32/95 (33.7%); 3: 10/95 (10.5%)
	P—0: 67/95 (70.5%); 1: 16/95 (16.8%); 2: 12/95 (12.6%)
PNS (n/N)	STAGE 1: 40/95 (42.1%); STAGE 2: 43/95 (45.3%); STAGE 3: 12/95 (12.6%)
AL (mm)	23.43 (22.84–24.88)
SE (D)	0.75 (−1.00–1.75)
CDE (%)	4.08 (2.95–5.33)
PHACO time (s)	23 (19–30)
IOLMaster^®^ 700 (PCB00)	22.50 (20.50–23.50)

LOCS = Lens Opacities Classification System; PNS = Pentacam^®^ Nucleus Staging; AL = axial length; SE = spherical equivalent; CDE = cumulative dissipated energy; PHACO time = total ultrasound time; n = number of patients with a certain stage; N = total number of patients. The table shows the distribution of general characteristics of the patients (N = 95) and parameters recorded during phacoemulsification. Age, AL, SE, CDE, PHACO time, IOL strength are shown as median and interquartile ranges (25th and 75th percentiles).

**Table 2 diagnostics-12-02449-t002:** Changes in pressure and visual acuity parameters.

	Before	1 Week After	1 Month After	3 Months After	6 Months After	*p*
**IOP (mmHg)**	14 (12–16)	13 (11–15)	12 (10–14)	11 (10–13)	11 (10–14)	**<0.001**
SBP (mmHg)	135 (145–140)	134 (124–140)	133 (124–140)	134 (125–139)	134 (125–139)	0.248
DBP (mmHg)	80 (75–90)	85 (77–88)	85 (79–90)	84 (79–89)	85 (79–90)	0.111
MAP (mmHg)	100 (93–104)	101 (93–104)	101 (94–106)	101 (95–104)	100 (93–106)	0.907
OPP (mmHg)	57.33 (51.39–60.44)	58.00 (54.22–61.00)	59.11 (54.50–62.22)	59.33 (55.11–61.56)	58.33 (54.50–61.22)	0.010
**BCVA (logMAR)**	0.16 (0.10–0.28)	0 (0–0.04)	0 (0–0.04)	0 (0–0.04)	0 (0–0.04)	**<0.001**

IOP = intraocular pressure; SBP = systolic blood pressure; DBP = diastolic blood pressure; MAP = mean arterial pressure; OPP = ocular perfusion pressure; BCVA = best corrected visual acuity. This table shows median and interquartile ranges for each parameter (25th and 75th percentile). *p* values and percentages of change presented were obtained 1 week after phacoemulsification. Friedman ANOVA test, significant difference (bold values) was found for values with *p* ˂ 0.001.

**Table 3 diagnostics-12-02449-t003:** Changes in vascular parameters in the corresponding layers at all time points.

	EA (mm^2^)	VA (mm^2^)	VPA (%)	TNJ	JD (Junctions/mm^2^)	TVL (mm)	AVL (mm)	TNEP	ML
Choroid	0.359	**<0.001**	**<0.001**	0.001	**<0.001**	0.319	0.839	**<0.001**	0.113
CC	0.742	0.279	0.243	0.834	0.789	0.914	0.008	**<0.001**	0.001
DVC	0.128	**<0.001**	**<0.001**	**<0.001**	**<0.001**	**<0.001**	**<0.001**	**<0.001**	**<0.001**
DCP	0.758	**<0.001**	**<0.001**	**<0.001**	**<0.001**	**<0.001**	**<0.001**	**<0.001**	**<0.001**
ICP	0.755	**<0.001**	**<0.001**	**<0.001**	**<0.001**	**<0.001**	**<0.001**	**<0.001**	**<0.001**
SVC	0.889	**<0.001**	**<0.001**	**<0.001**	**<0.001**	**<0.001**	**<0.001**	**<0.001**	**<0.001**
SVP	0.125	**<0.001**	**<0.001**	**<0.001**	**<0.001**	**<0.001**	**<0.001**	**<0.001**	**<0.001**
NFLVP	0.889	**<0.001**	**<0.001**	**<0.001**	**<0.001**	**<0.001**	**<0.001**	0.369	**<0.001**

CC = choriocapillaris; DVC = deep vascular complex; DCP = deep capillary plexus; ICP = intermediate capillary plexus; SVC = superficial vascular complex; NFLVP = nerve fiber layer vascular plexus; SVP = superficial vascular plexus; EA = explant area; VA = vessel area; VPA = vessel percentage area; TNJ = total number of junctions; JD = junction density; TVL = total vessel length; AVL = average vessel length; TNEP = total number of end points; ML = mean lacunarity. The table shows *p* values determined with Friedman ANOVA test, significant difference (bold values) was found for values with *p* ˂ 0.001.

**Table 4 diagnostics-12-02449-t004:** Percentage of detected changes in vascular parameters in the corresponding layers 1 week after phacoemulsification.

	EA (mm^2^)	VA (mm^2^)	VPA (%)	TNJ	JD (Iunctions/mm^2^)	TVL (mm)	AVL (mm)	TNEP	ML
Choroid	0.00%	**−2.14%**	**−2.14%**	1.32%	**1.34%**	0.08%	0.24%	**8.04%**	5.44%
CC	0.00%	0.57%	0.57%	−0.46%	−0.46%	−0.07%	35.57%	**−10.47%**	−4.99%
DVC	0.00%	**12.44%**	**12.44%**	**7.40%**	**7.38%**	**4.92%**	**184.83%**	**−47.79%**	**−27.24%**
DCP	0.00%	**12.75%**	**12.75%**	**11.77%**	**11.81%**	**8.04%**	**88.72%**	**−33.89%**	**−28.21%**
ICP	0.00%	**12.82%**	**12.81%**	**11.70%**	**11.69%**	**5.51%**	**93.70%**	**−37.65%**	**−28.22%**
SVC	0.00%	**20.00%**	**20.00%**	**21.08%**	**20.45%**	**11.07%**	**165.76%**	**−44.83%**	**−33.06%**
SVP	0.00%	**11.29%**	**10.83%**	**8.98%**	**8.99%**	**5.49%**	**53.13%**	**−29.58%**	**−20.97%**
NFLVP	0.00%	**36.37%**	**36.38%**	**63.43%**	**62.91%**	**33.62%**	**52.68%**	2.20%	**−37.12%**

CC = choriocapillaris; DVC = deep vascular complex; DCP = deep capillary plexus; ICP = intermediate capillary plexus; SVC = superficial vascular complex; NFLVP = nerve fiber layer vascular plexus; SVP = superficial vascular plexus; EA = explant area; VA = vessel area; VPA = vessel percentage area; TNJ = total number of junctions; JD = junction density; TVL = total vessel length; AVL = average vessel length; TNEP = total number of end points; ML = mean lacunarity. This table shows difference between values before and after surgery calculated as a percentage of change.

**Table 5 diagnostics-12-02449-t005:** Quality index of OCTA images before and after phacoemulsification.

OCT-A Image	Before	1 Week After	1 Month After	3 Months After	6 Months After	*p*
QI	35.4 (34.3–36.8)	35.4 (34.5–36.7)	35.4 (33.8–37.4)	35.4 (34.3–37.1)	35.5 (33.9–37.1)	0.742

OCTA = optical coherence tomography angiography; QI = quality index. This table shows median and interquartile ranges for quality index (25th and 75th percentiles). Friedman ANOVA test, the significance level was set to *p* < 0.001.

**Table 6 diagnostics-12-02449-t006:** Statistical analysis of changes in the surface of the foveal avascular zone.

Layer	Before FAZ Surface (mm^2^)	1 Week after FAZ Surface (mm^2^)	1 Month after FAZ Surface (mm^2^)	3 Months after FAZ Surface (mm^2^)	6 Months after FAZ Surface (mm^2^)	*p*	Change
Retina	0.326 (0.2539–0.4279)	0.2632 (0.1984–0.3446)	0.2539 (0.1993–0.3362)	0.2566 (0.1925–0.3198)	0.2554 (0.1815–0.3177)	**<0.001**	**−18.90%**

FAZ = foveal avascular zone. The table shows the median and interquartile ranges (25th and 75th percentiles). The percentage change was shown for values 1 week after phacoemulsification. *p* represents the total difference. Friedman ANOVA test, a statistically significant difference (bold values) was defined for values with *p* ˂ 0.001.

## Data Availability

The data presented in this study are available on request from the corresponding author.

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
