# Peer review of "Long-Term Functional Hyperemia after Uncomplicated Phacoemulsification: Benefits beyond Restoring Vision"

_diagnostics, 2022, doi:10.3390/diagnostics12102449_

Round 1

Reviewer 1 Report

The paper entitled “Long-term functional hyperemia after uncomplicated phacoemulsification: benefits beyond restoring vision” is a study long-term effects of uncomplicated phacoemulsification on macular perfusion, using optical coherence tomography angiography in patients undergoing phacoemulsification. The manuscript is of potential clinical interest.

The results indicate that phacoemulsification may have an effect on macular perfusion showing significant increases in various retinal vessel parameters using OCT. The hypothesis regarding induced functional hyperemia could provide physiological underlying retinal mechanisms to describe results.

The study provides objective results, which adds to current literature in this field. There are, however, several issues that need to be addressed by the authors, which include:

  1. The conclusions  need to be toned down considerably. The sentence stating that “the results advocate for earlier cataract surgery” is not justified. Although complication rates and risk of endophthalmitis related to this type of surgery are rather low, it is debatable whether or not a patient with good best corrected visual acuity should undergo early cataract surgery. The authors should reconsider this important issue and provide criteria that justify early surgery in healthy patients. Public hospitals with long waiting lists usually rely on functional cut-off values to determine if a patient is ready for surgery (i.e. BCVA < 5/10, anisometropia, etc.), thus promoting early cataract surgery could be problematic from a practical and ethical point of view.
  2. The statistical analysis for each parameter is important, however, 15 tables full of data render the paper difficult to read and disturb the flow of the manuscript. The authors should consider reporting these values in a different format or figure to avoid this exhaustive overhaul of data.    
  3. Considering this is a prospective study, the authors need to include the approval number and date of the local Ethics Committee in the Materials and Methods section.
  4. It would be interesting to compare the same prep and post-surgical parameters in patients with underlying ophthalmologic conditions, to determine if similar mechanisms are observed. This could be mentioned as future studies in the Discussion section.  
  5. The English can be improved for better flow.

Reviewer 2 Report

It is a very interesting and well designed study.

I have just minor remarks:

1.     Line 59-60: as quality index is device specific, please provide OCTA details of OCTA equipment

2.     Inclusion criteria: please emphasize that patients with retinal disorders that could affect OCT measurements were excluded ; was the exclusion made on the basis of pre-operative OCT ?

3.     Line 101 and further; please elaborate a little bit on the quality index as it is crucial for consistent OCTA measurements

4.     Lines 140-145 – were any OCTA scans/patients excluded due to variations in scan quality ?

5.     I would encourage authors to consider presenting just SCP and DCP values and skipping SVC and DVC or the opposite. It is really too much data for the reader.
